# Foliar Application of Different Iron Sources Improves Morpho-Physiological Traits and Nutritional Quality of Broad Bean Grown in Sandy Soil

**DOI:** 10.3390/plants11192599

**Published:** 2022-10-02

**Authors:** Abdel Wahab M. Mahmoud, Amira A. Ayad, Hend S. M. Abdel-Aziz, Leonard L. Williams, Rasha M. El-Shazoly, Ahmed Abdel-Wahab, Emad A. Abdeldaym

**Affiliations:** 1Plant Physiology Division, Department of Agricultural Botany, Faculty of Agriculture, Cairo University, Giza 12613, Egypt; 2Center for Excellence in Post-Harvest Technologies, North Carolina Agricultural and Technical State University, Kannapolis, NC 28081, USA; 3Department of Agricultural Botany, Faculty of Agriculture, Cairo University, Giza 12613, Egypt; 4Botany and Microbiology Department, Faculty of Science, New Valley University, Al-Kharja 72511, Egypt; 5Department of Vegetable, Faculty of Agriculture, Cairo University, Giza 12613, Egypt

**Keywords:** chelated iron, iron nanoparticles, sulphate iron, metabolic compounds, new reclaimed soil, phytohormones, plant growth, pod quality, *Vicia faba*

## Abstract

Nano-fertilizers are a new tool that can be used to address plant production challenges, and it addresses such nutrient deficiencies through smart agriculture approaches. Iron (Fe) is a vital element for several metabolic and physiological processes; however, Fe deficiency is common in poorly fertile soils (sand soil) and in arid areas. Therefore, additional research is required to select the most efficient form of iron absorbance. This research was implemented on broad bean plants (*Vicia faba* L. var. major Harz) to examine the impact of three iron sources: nano-iron (FeNPs, T1), iron sulfate (T2), and chelated iron (T3) as a foliar spray on the morphological properties, physiological attributes, and nutritional status of these plants compared to the untreated plants (control). The obtained results showed that foliar spraying with FeNPs, chelated iron and sulphate iron fertilizers increased plant height by 35.01%, 26.2, and 20.4%; leaf area by 38.8%, 18.3%, and 8.1%; the fresh weight of the plant by 47%, 32.8%, and 7.3%; the dry weight of the plant by 52.9%, 37.3%, and 11.2%; and the number of branches by 47%, 31.3%, and 25.6 %, respectively, compared to the control treatment (CT). Furthermore, the application of FeNPs, chelated iron, and sulphate iron fertilizers improved the number of pods by 47.9%, 24.8%, and 6.1%; the number of seeds by 32.8%, 7.9%, and 2.8%; and seed weight by 20.8%, 9.1%, and 5.4%, compared to control treatment (CT). Additionally, foliar application of FeNPs showed the highest values of photosynthesis rate (*Pn*), water-use efficiency (WUE), total chlorophyll, and phytohormones (IAA, GA3) compared to all the other treatments. The anatomical structure revealed an enhancement of leaf size and thickness (epidermis cells and mesophyll tissue) affected by FeNPs treatment compared to other treatments. Foliar application of FeNPs also improved the total content of carbohydrates, crude protein, element content (N, P, K, Ca, Na, Fe, Zn, Mn, and Cu), and some amino acids such as lysine, arginine, phenylalanine, isoleucine, and tyrosine in the seeds of broad beans. Based on the above results, the maximum values of all tested measurements were observed when FeNPs were used as the foliar spraying followed by chelated and sulphate iron fertilizers. Therefore, these findings suggest that using FeNPs, as a foliar treatment, could be a promising strategy for reducing the Fe deficiency in sandy soil and enhancing plant growth, pod yield, and pod quality of broad bean plants in addition to being environmentally favored in arid areas.

## 1. Introduction

Broad bean (*Vicia faba* L.) is one of the most important winter legume crops and a major source of protein for human nutrition [1]. It is the fifth most important crop, following pea, chickpea, common bean, and lentil. Broad bean (*V. faba* L.), also known as fava bean, field bean, or faba bean, is a type of bean belonging to the family *Fabaceae*. Broad bean’s nutritional value is largely due to its high-protein content, which often ranges from 25 to 35%. The bean is a good source of carbohydrates, minerals, and vitamins. This legume contains 50–60% carbohydrates, mainly starch, and 1–2.5% lipids, mostly oleic and linoleic acids. Globally, the cultivated area is around over 2.6 million hectares with an annual production reach of 5.4 million tons [2]. Hence, great attention was paid to the newly reclaimed soil outside the Nile Valley which represent about 3–4% of the total area of Egypt.

The newly reclaimed soil is mostly subjected to a variety of environmental exposures such as low-water availability, saline water, saline soil, soil alkalinity, nutrient deprivation, temperature changes, and high irradiances [3,4,5,6,7,8]. Therefore, when plants are exposed to low temperatures during various growth and development stages in the winter, they develop a variety of physiological and biochemical dysfunctions [9]. Thus, these physiological disorders caused a decline in growth as well as poor crop yields in terms of quality and quantity [10]. To increase plant tolerance, significant efforts must be made in this area. The application of optimal crop nutrition requirements is one of the most promising agricultural techniques. Foliar microelement supplementation is an important plant fertilization method. Foliar supplementation is more effective than soil application. Spraying nano-nutrients as fertilizer to minimize the consequences of plant deficiency in edible parts is considered to be a long-term solution. In the winter season, plants show several physiological and biochemical dysfunctions, when subjected to low temperatures during the different stages of growth and development [11].

Iron (Fe) is an important micro-nutrient for all organisms [12]. Iron deficiency is widespread in many crops. The Fe content of the soil is generally high, but a significant proportion is bound to soil particles [13,14]. Fe is usually in the form of insoluble Fe^3+,^ particularly in high-pH and aerobic soils; therefore, these soils are mainly deficient in the available form, Fe^2+^ [15]. Since plants absorb Fe^2+^ from the soil, Fe-deficient soils lead to Fe-deficient plants [16]. Plants use Fe in a variety of physiological processes, including respiration, chlorophyll biosynthesis, and redox reactions. Chlorophyll is a natural green pigment found in plants that plays a crucial role in photosynthesis [15,16,17]. Furthermore, Fe deficiency not only impacts the plant’s growth and development but can also lead to anemia in animals and humans [18]. Therefore, it is critical to increasing the effectiveness of Fe fertilizers use. Using iron in its traditional form is still the most popular method of increasing plant yields, but it is frequently ineffective because of poor nutrient-use performance. Further research is needed to determine the proper amount and the right form of iron to spray through foliar supplementation in order to decrease plant deficiency [19].

Chelated Fe (Che-Fe) has been suggested as an alternative method to enhance the plant’s ability to absorb iron. However, the application of chelated Fe to broad beans in various doses has received little attention to date, especially in tropical areas. Many studies have recently suggested using nanoparticles in the field of soil–plant nutrition in order to achieve sustainable development of agricultural production with less environmental risk [20]. Various studies have emphasized the value of using nanotechnology in agricultural production [21]. Therefore, several other nations such as the US, Japan, and Western Europe, have focused their research efforts on this area. As a result, nanomaterials will represent new perspectives on how to improve global agricultural production and will become the new agricultural development material [22]. Nanoparticles (NPs) are defined as materials with a very small diameter (between 0 and 100 nm) and a high specific surface area.

In comparison to the traditional materials, nanomaterials have several special functions resulting from the quantum size impact, dielectric confinement and macroscopic quantum tunneling as well as a few new functions [23]. NPs that contain magnetic materials made of elements like Fe, Co, and Ni, as well as their chemical compounds, are referred to as magnetic NPs (MNPs) [24,25]. Magnetic NPs (MNPs) are considered a group of NPs that contain magnetic materials of elements such as Fe, Co, and Ni, as well as their chemical compounds [25]. Due to their many applications, nanoparticles and their composites are currently attracting a lot of attention. They are extensively used in a variety of nanotechnology fields, including biomedicine, biotechnology, and agriculture such as fertilizers, fungicides, and bactericides in agriculture [25,26,27,28]. Numerous researchers reported that the application of nanoparticles (NPs) improved the growth, physiological activities, nutritional status, and yield of many economic crops [4,27,28]. Nonetheless, the effect of foliar spraying with nano-iron (FeNPs) as a replacement for conventional iron fertilizers on faba bean yield and its quality in newly reclaimed soils (sandy soil) is still unknown. Therefore, the aim of this study was to investigate the impacts of different iron sources chelated, conventional, and nanoforms, as a foliar application, on the growth performance, crop yield, and nutritional content of broad bean plants grown in newly reclaimed soils (sandy soil) with respect the environmental issues.

## 2. Results

### 2.1. Vegetative Growth

Our results indicated the impact of the three different iron sources (synthesized nano-iron (T1), chelated iron (T2), and iron sulfate (T3)) foliar spray on the broad bean (*V. faba* L.) growth parameters. The untreated broad bean plant was used as a control sample. The growth parameters were including the plant height, leaf area, number of branches, and fresh and dry weight of shoots (Table 1). Our results showed that the height of the plant treated with the synthesized nano-iron (T1) was significantly higher (120.6 ± 6.55, *p* ≤ 0.05) compared to the control sample, chelated iron (T2), and iron sulfate (T3) (78.4 ± 3.22, 106.3 ± 4.21, and 98.5 ± 2.55, respectively). Furthermore, the leaf area is an important indicator of plant growth. These findings revealed that T1 has a significant effect (20.46 ± 3.01, *p* ≤ 0.05) on the faba bean leaf area when compared to the control, T2, and T3 (12.52 ± 1.25, 15.33 ± 1.4, and 13.62 ± 2.34, respectively).

Additionally, these findings indicated that the foliar application of FeNPs (T1) significantly (*p* ≤ 0.05) enhanced the number of branches and fresh and dry weight of faba bean plants. Therefore, the number of branches per plant was significantly higher due to the use of Fe-NPs (6.8 ± 0.5, *p* ≤ 0.05) compared to the control sample, chelated iron, and iron sulfate treatments (3.2 ± 0.21, 5.1 ± 0.42, 4.3 ± 0.15, respectively). Similar results were observed in the fresh and dry weight of shoots. The highest values of shoot fresh and dry weight of plants were recorded in plants supplied with Fe-NPs compared to all the other treatments. The increment ratio reached 44.61%, 32.80%, and 10.58% for fresh weight and 46.99%, 31.27%, and 28.41% for dry weight of broad bean plants due to foliar application of magnetite iron nanoparticles (Fe-NPs), chelated iron, and iron sulfate, respectively.

### 2.2. Photosynthetic Pigments and Gas Exchange Parameters

As shown in Table 2, using different iron sources (synthesized nano-iron (T1), chelate iron (T2), and iron sulfate (T3)) as foliar application significantly affected (*p* ≤ 0.05) the total chlorophyll, carotenoids, photosynthesis rate (*Pn*), stomatal conductance (SC), and water-use efficiency (WUE) of the faba bean leaves. As a result of our findings, the content of chlorophyll in broad bean (*V*. *faba* L.) treated with T1 was significantly higher (101.12 ± 5.01, *p* ≤ 0.05) than in the control sample and the plants treated with T2 and T3 (86.49 ± 4.66, 90.45 ± 4.23, and 89.71 ± 3.99, respectively). Similarly, the concentration of the carotenoids in the plants exposed to the FeNPs was significantly higher (27.11 ± 2.02, *p* ≤ 0.05) than in the control sample (CT) and the plants exposed to the Fe-chelate and Fe-sulfate (22.19 ± 1.22, 24.79 ± 1.87, and 23.61 ± 3.11, respectively).

Likely, compared with the control, the greatest values of previous measurements were achieved when sprayed with the FeNPs. In addition, as a foliar application, Fe-chelate was statistically placed second after Fe-NPs in terms of photosynthetic pigments and photosynthesis apparatus, with the exception of carotenoid content in the faba bean leaves. Conversely, the lowest values of leaf total chlorophyll, carotenoids, photosynthesis rate, stomatal conductance, and water-use efficiency were typically observed in plants sprayed with water (control) and/or Fe-sulfate.

### 2.3. Leaf Phytohormones Concentration

As shown in Figure 1, spraying with different sources of iron fertilizers (Nano, chelate, and sulfate) significantly affected the concentration of indole-3-Acetic Acid (IAA), gibberellic acid (GA3), and Abscisic acid (AB) in the leaves of broad bean plants. The concentration of indole-3-Acetic Acid (IAA) significantly increased in the leaves of broad bean plants treated with various ions (Nano, chelate, and sulfate) compared to the control. The highest value of IAA was detected in the leaves of plants treated with Fe-NPs compared to all the other treatments. Furthermore, both iron fertilizers (Fe-chelate and Fe-sulfate) also enhanced the concentration of IAA in broad bean leaves, but their effects were lower than the Fe-NPs treatment in comparison to the control treatment. Compared to the control, the enhancement ratio in the leaf IAA concentration was 37.34%, 16.02%, and 16% for Fe-NPs, Fe-chelate, and Fe-sulfate treatments, respectively.

Compared to the corresponding control, foliar application of Fe-NPs enhanced the gibberellic acid (GA3) concentration in the leaves of broad bean plants by 29.41%. Additionally, both iron fertilizers, chelated iron, and iron sulfate did not show any significant changes in GA3 concentration. The abscisic acid concentration (ABA) in plants was significantly lower in Fe-NPs and Fe-chelate treatments than in Fe-sulfate and control treatments. The reduction ratio in the ABA concentration in the leaves of plants treated with Fe-NPs and Fe-chelate treatments was 44% and 32%, respectively, compared to the control treatment.

### 2.4. Seed Nutrient Content

Data in Table 3 shows that the different sources of applied iron (T1 = Nano, T2 = chelate, and T3 = sulfate) significantly increased (*p* ≤ 0.05) the content of macro-elements (N, P, K, and Ca) in leaves of broad bean plants (*p* ≤ 0.05). Compared to the control treatment, the maximum values of macro-nutrient content (N, P, K, and Ca) were recorded in plants treated with Fe-NPs treatment (T1) followed by Fe-chelated (T2) and Fe-sulfate treatments (T3). Furthermore, the Fe-NPs treatment increased nitrogen and phosphorus content by about two folds of the control, while potassium and calcium content in seed was enhanced by about four folds of control.

Similar significance was observed in the micro-element content (Fe, Zn, and Mn) in broad bean seeds (*p* ≤ 0.05). The different applied treatments enhanced the concentrations of seed Fe, Zn, and Mn content more than the control treatment (CT). The greatest concentration of Fe, Zn, and Mn in seed was observed in broad bean plants treated with Fe-NPs (T1) in comparison to control. Both Fe-chelate and Fe-sulfate increased the content of microelements in broad bean seeds, but less than Fe-NPs, compared to control (*p* ≤ 0.05). The Fe and Zn contents peaked up to about 2.5 and 2 folds of the control, respectively.

### 2.5. Seed Carbohydrate, Crude Protein, Fat and Moisture Content

Results presented in Table 4 showed the impact of the different sources of iron on the carbohydrate, crude protein, fat, and moisture content in the broad bean seeds. Applying different forms of iron (Nano, chelate, and sulfate) as a foliar application, improved the total carbohydrate content in the seeds of broad bean. Compared to the untreated plants (CT = 493.29 ± 23.18), the total carbohydrate content in seeds of broad bean was significantly increased (619.48 ± 34.12, *p* ˂ 0.05) by applying Fe-NPs treatment. Another significant increase was observed when comparing T1 treatment with T2 and T3 (546.73 ± 32.78, and 521.18 ± 44.02, respectively). Moreover, treating the broad bean plants with magnetite nanoparticles and chelated iron (Fe-NPs and Fe-chelate) as foliar fertilizers significantly increased protein percent (35.14 ± 3.11 and 30.25 ± 2.99, respectively) compared to the control sample (26.30 ± 2.21). Furthermore, compared to CT treatment, T1 treatment increased the fat and moisture content in the broad bean seeds (2.03 ± 0.12 and 8.49 ± 0.51, respectively). However, the other treatments, T2 and T3, had no effect on the fat or moisture content of broad bean seeds.

### 2.6. Amino Acid Compounds Profile of Broad Bean Seeds

Essential amino acid composition (threonine, valine, leucine, and histidine acids) in seeds of broad bean plants treated with various Fe foliar fertilizers (nanoparticles, chelated iron, and iron sulfate) is shown in Figure 2. Spraying different types of iron fertilizers on the plants significantly increased the content of the plant amino acid. Furthermore, on average the T1 treatment (Fe-NPs) yielded the best results. However, a significant increase in arginine and lysine was only observed in the nanoparticles treatment (T1), with no statistical difference in arginine and lysine content under chelated iron or iron sulfate. Compared with CT treatment, the maximum value of phenylalanine content was recorded in seeds of plants treated with FeNPs (T1) followed by iron sulfate (T3), whereas chelated iron (T2) had no effect on phenylalanine content see Figure 3.

Data of non-essential amino acid contents were shown in Figure 3. The amino acids arginine and lysine significantly increased due to FeNPs treatment (T1), whereas arginine and lysine contents did not show any difference when treated with chelated iron (T2) or iron sulfate (T3). In contrast, the amino acid aspartic did not statistically differ between the magnetite nanoparticle treatment and the control treatment. Aspartic acid was significantly raised by iron sulfate treatment (T3) and then iron chelate therapy (Figure 3). The amino acid proline was not statistically different in the presence of chelated iron or magnetite nanoparticles. However, proline content in broad bean seeds was considerably reduced in the presence of iron sulfate when compared to the untreated plants (CT).

Results presented in Figure 3 indicated that the iron sulfate treatment had no effect on the alanine amount in the broad bean seeds. However, FeNPs (T1) and chelated iron (T2) treatments revealed a much lower level of alanine. Furthermore, all the broad bean plants treated with the three Fe foliar fertilizers (magnetite nanoparticles, chelated iron, and iron sulfate) treatments had considerably less serine and glutamic acid in their seeds when compared to the control treatment (CT).

### 2.7. Yield and its components

All the yield parameters, including the number of pods per plant, the number of seeds per pod, the harvest index, the biological yield, and the weight of 100 seeds, significantly changed in response to the treatments in the study (Table 5), though the degree of response varied depending on the parameter. Additionally, compared to iron sulfate and Fe-chelated treatments, Fe-NPs achieved a faster rate of change in such yield metrics. Furthermore, the foliar spray with Fe-NP solutions greatly improved all yield and component parameters when compared to the control and other iron sources. Our findings revealed that the broad bean seeds treated with a spray solution of Fe-NPs followed by Fe-chelated compared to iron sulfate and control, the maximum number of pods, number of seeds, the weight of 100 seeds, and harvest index were recorded. Our results indicated that there were no significant differences between the iron sulfate and control treatments. On the contrary, untreated plants (control) and plants sprayed with iron sulfate solution had the highest biological yield (Table 5).

### 2.8. Anatomy of the Leaflet

Table 6 shows the microscopical counts and measurements of specific histological characters in transverse sections through the first leaflet blade of the compound leaf developed on the median portion of the main stem of broad bean control plants and those treated with synthesized iron nanoparticles (FeNPs) about 6 nm in size. Figure 4A,B and Table 6 show microphotographs demonstrating the effect of this treatment. Counts and measurements of specific histological features in transverse sections through the first leaflet blade of the compound leaf formed on the median portion of the main stem of the broad bean plant as influenced by exogenous application of Fe-NPs are given in microns (µ).

Our results demonstrated that the broad bean leaflet blade thickness was raised by 59.7 and 45.2%, respectively, when treated with synthetic iron nanoparticles (FeNPs) at a concentration of 6 nm in size compared to control plants (CT). It is obvious that the palisade and spongy tissues both significantly increased in thickness when compared to the control plants, by 56.3 and 44.4%, respectively, explaining the increase in lamina thickness. Similarly, treatment with iron nanoparticles caused the midvein bundle’s primary vascular bundle to noticeably grow in size (FeNPs, T1).

### 2.9. Correlation Study

After spraying the broad bean plants with several types of iron fertilizers, the changes in agrochemical and biochemical features of the broad bean plants and seeds are demonstrated using principal component analysis (PCA), a heatmap (Figure 5 and Figure 6) and Pearson’s correlation (Appendix A). In this study, the use of the XLSTAT program and two-dimensional principal component analysis (PCA) was performed. A total of 27 wide bean metrics were combined considering the changes in the plant growth metrics and biochemical features of broad bean plants sprayed with various types of iron. The results of agronomical and biochemical indicators of broad bean plants were further integrated and analyzed using PCA. Principal components (PCs) gave 96.51% of the total alteration of the data set. The contribution rate of PC1 and PC2 was 92.43% and 4.08% of the variance in the data set, respectively. PCA and Pearson’s correlation showed that the biological yield correlated positively with plant height, leaf area, number of branches/plant, fresh and dry shoots, number of seed/pod, and number of pods. A similar relationship was observed between biological yield and total chlorophyll, carotenoids, photosynthesis rate (*Pn*), stomatal conductance (SC), water use efficiency (WUE), and phytohormones (GA3, IAA). Furthermore, the content of Fe, total carbohydrates, and crude protein in the seed was positively associated with total chlorophyll and biological yield.

Furthermore, foliar spraying with diverse sources of iron fertilizers to *V*. *faba* plants induced variations in their seed amino acid profiles. As shown in Figure 5, the heatmap revealed that the spraying with diverse sources of iron fertilizers was correlated positively with upgrading some essential and nonessential amino acid compounds in the seeds of *V*. *faba* plants, particularly Threonine, Valine, Isoleucine, Leucine, Phenylalanine, Histidine, Lysine, and Glycine. However, the tyrosine and aspartic compounds increased with the foliar addition of Fe-sulfate fertilizer. On the other hand, foliar addition of Fe-sulfate to *V. faba* plants induced the reduction in Lysine, serine, and proline acid in seeds.

## 3. Discussion

Iron (Fe) is one of the most essential microelements which has a significant impact on a plant’s health, yield quality, and the production of by-products in many crops [29]. Due to the fact that Fe is necessary for a variety of physiological functions in plants, including the production of DNA, respiration, photosynthesis, and protein. In contrast, a lack of Fe in Egyptian soils is strongly associated with a variety of soil characteristics such as high pH, high salinity, low organic matter, and free calcium carbonate [30]. Nano fertilizers might be used to enhance traditional agricultural practices and furnish sustainable development by decreasing agricultural input wastes and enhancing management and preservation strategies [31,32].

The current study showed that the maximum values of vegetative growth measurements, including plant height, leaf area, number of branches per plant, shoot fresh weight, and shoot dry weight, were observed in the plants treated with nano-iron, followed by chelated iron and iron sulfate, compared to unfertilized plants (Table 1). While the highest values of the aforementioned parameters were noted in plants treated with FeNPs treatment. Furthermore, the current study also indicated that using FeNPs showed enhancement in the leaf size and thickness of epidermis cells and mesophyll tissue (Table 6 and Figure 4). A positive correlation was observed between Fe in seeds of broad beans and the previous plant growth parameters (Figure 5 and Appendix A). The improvement of vegetative growth parameters could be due to increasing the amount of chlorophyll, the rate of photosynthesis, and nutrient uptake, all of which significantly increase the accumulation of polysaccharides and organic matter in various plant organs (Table 2 and Table 3). These findings were consistent with the results presented by Elfeky et al., [33] that increasing the concentrations of Fe_3_O_4_ NPs (1, 2, and 3 mg/L) increased total plant mass, root length, number of leaves, and weight for *Ocimum basilicum* L. Another study claimed that ryegrass and pumpkin plants both showed an improvement in root elongation [34]. Moreover, El-Gioushy et al. and Plaksenkova et al. [35,36] claimed that spraying nano-iron significantly improves plant-growth parameters and fruit quality of over-chelated and conventional iron treatments due to increasing the concentration of the photosynthesis pigments (chlorophyll and carotenoid) and nutrient absorption.

Chlorophyll is the essential pigment in photosynthesis and it is responsible for absorbing and transferring light energy [37]. The content of leaf chlorophyll and carotenoid is the main indicator to describe the physiological performance of plant photosynthetic tissues, which considerably influenced plant photosynthesis [38,39]. Carotenoids are the bioactive pigment, as antioxidants, that protect the chlorophyll from photodegradation and oxidation [40,41]. Results of the present study showed that spraying with different sources of iron fertilizers significantly increased the content of leaf chlorophyll by 14.47%, 4.38%, and 3.59% and leaf carotenoid by 18.15%, 10.49%, and 6.01%, respectively, compared to the untreated plants. The superiority of leaf chlorophyll and carotenoid contents was noted in the broad beans treated with FeNPs. One reason was that FeNPs boosted the activity of iron oxygen reductase, indirectly enhancing porphyrin metabolism to produce aminolevulinic acid, a precursor to chlorophyll [42]. In agreement, Mahmoud et al. [43] reported that FeNPs and ZnNPs significantly maximize the leaf carotenoid content in the red radish. A similar finding by Plaksenkova et al. [35] was observed, that the FeNPs increased the chlorophyll level in the leaves of *Eruca sativa* compared to that of the control group. Furthermore, it is critical to encourage chlorophyll concentration in plants for biotechnological applications [44].

As shown in Table 2, the foliar application with the different iron fertilizers had a positive effect on the gas exchange parameters, i.e., photosynthesis rate, stomatal conductance, and water-use efficiency (WUE). Compared to untreated plants, the foliar application of FeNPs, chelate iron, and iron sulfate increased the leaf photosynthesis rate (*Pn*) by 57.97%, 40.57%, and 17.22%; stomatal conductance (Sc) by 40%, 27.6%, and 16%; and water-use efficiency (WUE) by 49.56%, 24.65%, and 14.62%; respectively. In addition, the photosynthesis rate is positively associated with the chlorophyll content (Figure 5 and Appendix A), which suggests that FeNPs can induce the enhancement of photosynthesis more than other applied iron fertilizers by increasing the chlorophyll content. The improved photosynthetic activity could be attributed to the role of iron indirectly in enhancing the size and yield of the chloroplast and the rubisco protein content [41,45]. The increase in photosynthesis rate (*Pn*) was expected to increase the water-use efficiency (WUE) of plants [4,39,46,47]. The photosynthesis rate correlated positively with stomatal conductance (*SC*) and WUE (Figure 5 and Appendix A). As for stomatal conductance, El-Gioushy et al. [36] reported that FeNPs fertilizer significantly increased the accumulation of Fe in plant leaves, this element is helpful in changing stomatal conductance, net carbon dioxide (CO_2_), and fixation rate. Lui et al. [46] stated that the accumulation of Fe in leaves causes an improvement in stomatal opening and thereby increases CO_2_ uptake and enzymatic activity at the chloroplast which consequently improves the photosynthesis efficiency. Several studies have displayed a similar response of FeNPs on different crops in this respect [36]. The current results were in accordance with those stated by Kazemi [48], who showed that the foliar application of FeNPs significantly improved the *Pn,* WUS, and stomatal conductance (SC) of cucumber plants, and Fe concentration was increased too. Furthermore, the untreated plants resulted in lower *Pn*, SC, and WUF (Table 5). Fernández et al. [49] reported that lower Fe in plant leaves causes a reduction in photosynthetic rates, stomatal apertures, and transpiration rates.

Phytohormones are bio-constituents produced in trace quantities by plants. They play vital roles as signaling molecules to stimulate various aspects of crop growth and development [50]. The ABA hinders growth, induces senescence, and reduces metabolism. Compared with the control, the minimum concentration of leaf ABA content was recorded in broad bean plants sprayed with FeNPs followed by chelated iron treatments (Figure 1). Pearson’s correlation shows a negative link between the leaf ABA content and seed Fe content (Appendix A). In earlier studies, the ABA concentration of plants significantly increased under Fe deficiency [51]. Lower ABA concentration can induce normal growth and delay aging. Other plant hormones (e.g., IAA and GA) simulate crop growth and postpone senescence. Many previous studies have shown that the ABA content increased, and the IAA and GA contents decreased under environmental stress conditions [50,51]. Our results were consistent with these findings; that is, the contents of IAA and GA3 were generally highest in FeNPs treatment than in other treatments. The correlation study (PCA and Pearson’s correlation) showed that the seed Fe content positively correlated with leaf IAA and GA3 (Figure 5 and Appendix A).

Our results were in agreement with the findings of El-Gioushy et al. [36] the content of essential elements (N, P, K, Ca, Fe, and Mn) in the seeds of broad bean plants was significantly higher after receiving foliar applications of various types of iron than untreated plants. In agreement with the results from Vattani et al. [52] which reported that applying nano-iron fertilizer to two varieties of spinach improves the accumulation of iron and potassium. Similar results founded by El-Gioushy et al. [36], who stated that the maximum accumulation of N, P, K, Mg, Ca, Fe, and Mn were in leaves and fruits of orange trees sprayed with nano-iron (FeNPs) followed by chelated iron and conventional iron (FeSO_4_), compared with control. Regarding the leaf Zn content, our result showed that the highest content of leaf Zn was observed in faba bean plants treated with FeNPs compared to all the treatments. Furthermore, seed Fe content correlated positively with seed Zn content (Appendix A). Similar results were observed by El-Gioushy et al. [36] and Rasht [53] who reported that the foliar application of nanoform of iron fertilizers increased the content of Zn in leaves and economic parts of plants. This improvement in the accumulation of essential nutrients N, P, K, Mg, Ca, Fe, Zn, and Mn in plant leaves and edible parts could be due to the increasing root growth of broad bean plants sprayed with different sources of iron, and thus, improved the absorption of nutrients by the plant [54,55]. Whereas, Konate et al. [54] illustrated that FeNPs might interact with plants to produce OH free radicals, which can stimulate the degradation of pectin in the plant cell wall and relax the root cell wall to increase plant root growth. In corn (*Zea mays* L.), FeNPs (particle size 17.7 nm) at 20 mg/L under hydroponic conditions significantly increased root elongation [47]. Generally, the increase in the accumulation of Fe element in the leaves of treated broad bean plants with nano-iron fertilizers than other applied fertilizers may be linked to the properties of Fe nanoparticles which are characterized by a smaller surface area, higher absorption, and more ease of attachment than other forms of iron [56,57].

In order to determine the most effective form of iron (Fe) for the plant’s nutrition compared with the synthesized nano-iron, chelated iron, and iron sulfate. Foliar application of Fe forms was employed and suggested to treat the Fe deficiency in broad bean plants [36,58,59,60]. This might explain why we compared the nano, chelated, and conventional iron to explore the optimum form in terms of efficiency in plant nutrition [58], foliar application of Fe forms was used and recommended to correct the Fe deficiency in broad bean plants. Spraying faba bean plants with nano-iron, chelated iron, and iron sulfate improved total carbohydrate by 20.37%, 9.77%, and 5.35% and crude protein by 25.2%, 13.06%, and 7.43%, respectively, compared to the untreated plants. In addition, Barłóg et al. [59] stated that the nutritional value of crude proteins is based mainly on the capacity to satisfy the body’s requirements from nitrogen and essential amino acids. The findings of the current study exhibited that the amino acid composition was significantly influenced by the foliar application of different sources of iron fertilizers. The FeNPs treatment showed the highest content of lysine, arginine, phenylalanine, isoleucine, and tyrosine in the seeds of faba beans (Figure 2, Figure 3 and Figure 6).

Various studies suggested that FeNPs perform a crucial role as a cofactor for enzyme activation and reaction and have been found to be an efficient strategy to increase leaf chlorophyll content, photosynthetic rate, carbon absorption, and nitrogen metabolism [47,58]. Furthermore, the PCA and Pearson’s correlation show a significant association between the seed Fe content, crude protein, and total carbohydrate contents (Figure 5 and Appendix A). Nadi et al. [61] reported that spraying plants of Faba bean with FeNPs significantly improves crude protein levels. The protein levels may have increased as a result of the function FeNPs play in photosynthesis, DNA transcription, RNA synthesis, and enzyme reactions, which all contribute to growth and development. El-Gioushy et al. [36].

Foliar application of different iron sources (synthesized nano-iron, chelated iron, and iron sulfate) significantly enhanced the pod yields and its components. Additionally, there is a favorable correlation between the seed’s iron content and biological yield. Similar correlations were discovered between biological output and morphological growth (including leaf area, plant height, the fresh and dried weight of shoots, as well as physiological characteristics (*Pn*, WUE, SC, carbohydrate, and crude protein content). The highest improvement in the number of pods/plants, number of seed/pods, biological yield, and weight of 100 seeds was observed in broad bean plants treated with FeNPs fertilizer more than in other treatments. While Fe serves as an electron carrier in respiration and photosynthesis, in the production and detoxification of free oxygen (O_2_) radicals, O_2_ transport, and lowering of the molar concentration of oxygen [3,4,5,6,7,8,15,16,17]. Our findings show that FeNPs have a great impact on introducing more soluble Fe, resulting in more surface area for the broad bean plant’s metabolic reactions and an improvement in vegetative growth, photosynthesis rates, metabolic compound accumulation (carbohydrate, fats, and crude protein), and yield (Table 1, Table 2 and Table 5).

The results from Nadi et al. [61] stated that the application of FeNPs in optimum concentration had a positive impact on fava bean yield and yield characteristics. These results show that Nano-Fe fertilizers are more efficient or effective than Che-Fe and Conv-Fe fertilizers, as they improved plant growth and increased metabolic efficiency such as photosynthesis, which leads to higher photosynthates accumulation and translocation to the plant economic parts and total yield. Hence, choosing the right form and concentration of iron is critical for receiving the most benefits. Since the excess concentration of FeNPs caused a serious reduction in the growth and production of plant crops due to the formation of reactive oxygen species (ROS) which are responsible for cell damage [27]. Therefore, many researchers have stated that using FeNPs at the optimal concentration significantly improved vegetative growth characteristics, pod quantity, and nutritional status of crops in low fertile sandy soil and arid regions [36].

## 4. Materials and Methods

Two field experiments at a private farm during the winter seasons 2020–2021 and 2021–2022, in Wadi El-Natron El-Behera governorate, Egypt were conducted (longitude: 2540 E; latitude: 28,200 N; and altitude:125 m). Three different treatments of iron form (T1) synthesized nano-iron (FeNPs), (T2) chelated iron (Fe-chelate), and (T3) iron sulfate (Fe-sulfate) were used. Water spray was used as a control. These treatments were sprayed at a rate of 100 mg.g^−1^ after 60, 90, and 120 days from seed sowing. Seeds of broad bean (*Vicia faba* L. var. major Harz) were donated by Field Crops Research Institute, Agricultural Research Center (ARC), Giza, Egypt. *Rhizobium leguminosarum* strain was inoculated into the Broad seeds before sowing. The experimental field’s soil was carefully prepared and divided into plots that measured 4 × 2.5 m. Each plot contained four rows, with 50 cm space between each row and 30 cm between the plants. Fifteen days before planting, the compost was obtained from the Soil, Water and Environment Research Institute, and it was applied at a rate of 5 ton/feddan. The chemical analysis of applied compost was shown in Table 7. The plant pods were harvested after 98 days. The physicochemical and microbial properties of the experimental soil were investigated (Table 8).

### 4.1. Synthesis and Characterization of Iron Nanoparticle

The Fe_3_O_4_ magnetic nanoparticles were prepared by following the protocol as per Hong et al. [62] with slight modifications. The co-precipitation of Fe^3+^ and Fe^2+^ at a molar ratio of 3:2 with aqueous ammonia (0.3 mol.L^−1^) was prepared as precipitating agent. The size and shape of Fe_3_O_4_ magnetic nanoparticles were observed by using transmission electron microscopy (TEM) using an electron acceleration voltage of 60 kV (Figure 7). The TEM samples were prepared by dropping a few drops of the solution on a carbon-coated copper grid (Okenshoji Co., Ltd., microgrid B).

### 4.2. Growth and Yield Parameters

Plants samples were selected randomly from each plot to measure the following growth parameters; plant height (cm), leaf area (cm^2^), number of branches plant, shoot fresh and dry weights (g.plant^−1^), number of pods, pod weight, 100-seed weight (g), and seed yield (kg/Fad.). The biological yield (ton/ha) and harvest index (%) were assessed according to the method reported by Abdalla et al. [63].

#### 4.2.1. Physiological Traits

##### Photosynthetic Pigments

The photosynthetic pigments in leaves, including chlorophyll and carotenoids, were extracted and quantified based on the method of Merwad et al. [64] with a few modifications. Approximately 20 mg of frozen leaves were immersed in 2 mL of acetone (80%, Sigma-Aldrich Co. LLC). Then, the extraction was centrifuged at 3000 rpm for 15 min. The absorbance of leaves extracts supernatants were measured by spectrophotometer (Helios UVG1702E, England) at wavelengths (λ) 663-nm and 647-nm to determine total chlorophyll content and at 470-nm [65] to assay the total carotenoid content and following equations.

-Chlorophyll a = 12.70 × A_663_ − 2.79 × A_647_-Chlorophyll b = 20 × A_647_ − 4.62 × A_663_-Carotenoids = {1000 × A470 − (3.27 chl. A − 104 chl. B)}/229

##### Gas Exchange Parameters

The fifth leaf of five randomized plants was selected from each treatment to determine determined leaf net photosynthesis (*Pn*) and stomatal conductance (SC) using. A portable Li-Cor 6400 (LiCor, Lincoln, NE, USA) equipment with an infrared gas analyzer. These measurements were taken in mid-morning (11:30 am–1:30 pm) under the following environmental conditions, with a temperature of the leaf chamber around 25.5–27.8 °C, external CO_2_ (ca. 399 mg.Kg^−1^), a light intensity of about 12500 µmol m^−2^ s^−1^, and relative humidity was about 69%. The water use efficiency (WUE) was calculated as the rate of photosynthesis divided by the rate of transpiration [66,67].

##### Seed Nutrient Content

Broad bean seeds were freshly cleaned, washed, weighed, and then put in an oven until a constant dry weight. The fresh and the dry weights were recorded. A 5 g of dried seeds from each treatment were grounded using an electric mill. The grounded samples were digested with sulfuric acid and perchloric acid (3:1) and used for chemical analysis. The percentage of nitrogen, phosphorus, potassium, and calcium in dry-plant samples was determined according to the method described by A.O.A.C. [68]. The nitrogen content of the samples was determined using the Kjeldahl method. Phosphorous was determined using ammonium molybdate and ascorbic acid [69]. Potassium was determined by a flame photometer, while Fe, Zn, Mn, and Ca were determined using Atomic absorption spectroscopy [70].

##### Gibberellic Acid, Indole-3-Acetic Acid and Abscisic Acid

The quantification of phytohormones content in broad bean leaves was performed according to the protocol from Abdelaziz et al. [3], with slight modifications. Freeze-dried broad bean leaves were milled into a fine powder. A 10 mg of the leaves powder was washed four times with 80% methanol (*w/v*%) and 2,6-bis (1,1-dimethylethyl)-4-methylphenol at 5 °C in the dark room. The mixture was centrifuged at 4500 rpm for 4 min. The supernatant was discarded, and the pellet was mixed with an equal volume of ethyl acetate. The solution was thoroughly mixed and then filtered using Whatman filter paper no. 2. Quantification of abscisic acid (ABA), Gibberellic acid (GA3), and indole-acetic acid (IAA) were performed by using Ati-Unicum gas–liquid chromatography. The values of phytohormones were expressed as ma.100 g^−1^ fresh weight.

##### Seed Protein Content

Crude protein content in the broad bean seed was measured using the Kjeldahl method according to A.O.A.C. [71]. Approximately 0.5 g of dried seed was hydrolyzed with 5 mL concentrated sulfuric acid (H_2_SO_4_) containing perchloric acid (HClO₄) in a heat block at 420 °C for 2 h. After cooling, distilled water was added gentility to the hydrolysates before titration. Crude protein in the seeds was calculated by multiplying the values of total nitrogen in the conversion factor of 6.25 [72].

##### Seed Carbohydrate and Fat Content

The total carbohydrates of broad beans were determined using phosphomolybdic acid method according to AOAC [68]. One gram of ground bean seeds samples was transferred into tubes, and then 0.2 mL of ethanol (80%) and then 3 mL of thermostable amylase were added. The mixture was incubated in a water bath min at 95 °C for six minutes. Then, the mixture was transferred into a 50 °C water bath, and then 4 mL of (200 mM) C_2_H_3_NaO_2_ (sodium acetate) was added. The mixture was centrifuged at 13,000 rpm for four minutes, and a 0.1-mL aliquot of each sample was inserted into the test tube and incubated for 20 min in a dark room. The samples of carbohydrate content were measured by using the spectrophotometer (Helios UVG1702E, England) at wavelengths 630-nm and the results were expressed as mg.g^−1^ DW.

The fat content in the seeds of broad beans was assessed using the Soxhlet extraction according to the AOAC method [68]. Three grams of dry seeds sample was mixed with a buffer solution containing petroleum ether and methanol ratio of (90:10, *v/v*) at 60–80 °C for six hours. After that, the mixture was dried in a forced air oven at 100 °C for one hour and then cooled down in a desiccator for 30 min. The total fat content was calculated by subtracting the difference in the weight of the vessel containing fat and the empty vessel.

##### Seed Amino Acids Content

The amino acid compositions of seeds were determined using the amino acid analyzer apparatus model (LC 3000 Eppendorf, Central Lab. of Desert Research Center). Hydrolysis was performed according to the technique stated by Pellet and Young [73].

#### 4.2.2. Leaf Anatomy

The broad bean (*V. faba* L.) leaves were collected from the mature leaves on the shoot during vegetative growth. The specimens were taken from the leaf between mid-vein and the leaf margin and then fixed in formalin-acetic acid alcohol (FAA) using 70% ethanol. The specimens were gradually dehydrated in a tert-butyl alcohol (TBA) series [74] and embedded in paraffin wax (m.p. 56 °C). Sections were cut on a rotary microtome at a thickness of 8–10 μm (Model RM2245, Leica Microsystems). Paraffin was removed with xylol solvent and slides were stained with safranin FCF methanol and fast green, and then mounted in Canada balsam [74]. The selected sections were examined and photographed using a light microscope (Model BX51; Olympus Optical).

### 4.3. Statistical Analysis

The data obtained during the two seasons in this study were subjected to combined analysis after doing the normality distribution test [75] and homogeneity test [76]. In addition, the combined data were subjected to the statistical analysis of variance. The means were compared according to the Duncan multiple comparison test at the 5% level of significance using the SPSS statistical 11.0 package. Moreover, principal component analysis (PCA), and clustered correlation heatmaps among parameters were performed using the XLSTAT program. Principal component analysis (PCA) and Pearson’s correlation are multivariate statistical approaches used to simplify a large number of variables into a small number of principal components (or factors) based on the association patterns of the original variables. The XLSTAT is defined as a statistical program that can be used to conduct multivariate analysis on large data sets. All data are stated as means ± standard error of the values achieved by six independent measurements.

## 5. Conclusions

The findings of the current study reported that the foliar application of broad bean plants grown under sandy soil and similar agricultural practices with Fe-NPs is a beneficial technique for improving morphological performance, physiological traits, nutritional status, pod yield, and pod quality compared with chelated and sulfate iron fertilizers. These variations are due to the variable uptake ratio of Fe from the various forms of iron fertilizers used. Therefore, nanoform of Fe is suggested in new reclaimed soil (sand soil) and arid areas to attain higher pod yield and pod quality. Moreover, further research is needed to investigate the impact of different levels of iron nanoparticles on oxidative stress and antioxidant compounds in plants.

## Figures and Tables

**Figure 1 plants-11-02599-f001:**
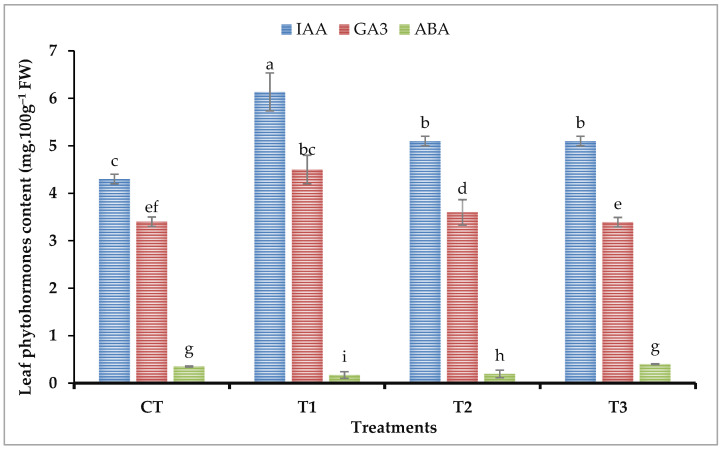
Impact of diverse iron sources, as foliar spray, on Leaf phytohormonal content of *Vicia faba* plants. Means of each column followed by the same letter are not statistically different according to the Duncan multiple comparison test at the 5% level. Each value is the average of 6 replicates over two seasons. Bars indicated to standard error (±SE). T1 = Synthesized nano-iron, T2 = Chelated iron, T3 = Iron sulfate, and CT = Control. FW = fresh weight.

**Figure 2 plants-11-02599-f002:**
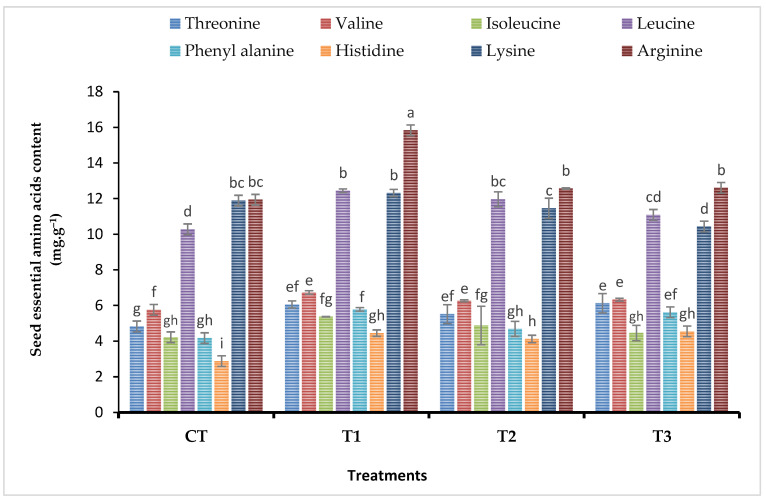
Impact of diverse iron sources, as foliar spray, on seed essential amino acid content of *Vicia faba* plants. Means of each column followed by the same letter are not statistically different according to the Duncan multiple comparison test at the 5% level. Each value is the average of 6 replicates over two seasons. Bars indicated to standard error (±SE). T1 = Synthesized nano-iron, T2 = Chelated iron, T3 = Iron sulfate, and CT = Control.

**Figure 3 plants-11-02599-f003:**
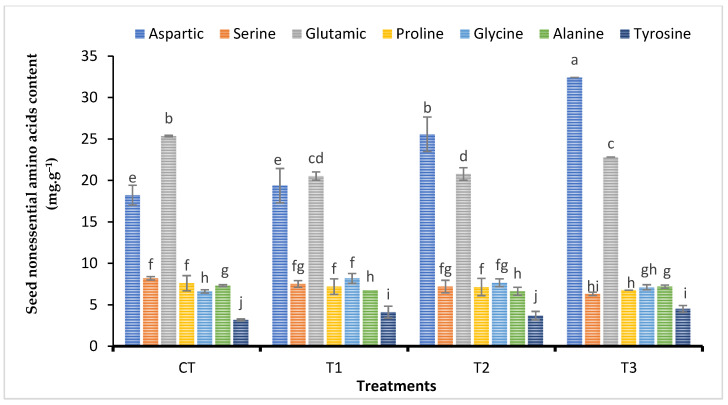
Impact of diverse iron sources, as a foliar spray, on seed nonessential amino acid content of *Vicia faba* plants. Means of each column followed by the same letter are not statistically different according to the Duncan multiple comparison test at the 5% level. Each value is the average of 6 replicates over two seasons. Bars indicated to standard error (±SE). T1 = Synthesized nano-iron, T2 = Chelated iron, T3 = Iron sulfate, and CT = Control.

**Figure 4 plants-11-02599-f004:**
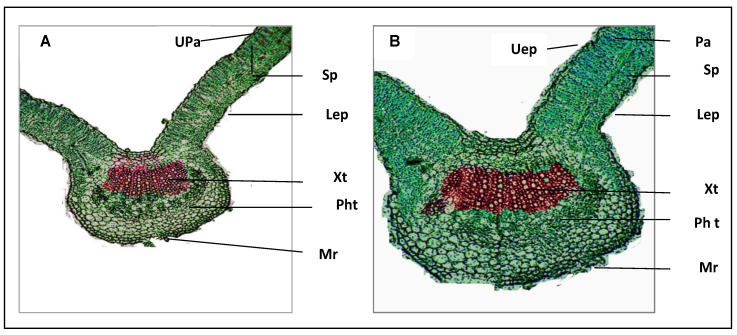
Transverse section through the first leaflet blade of the compound leaf developed on the median portion of the main stem of broad bean plant as affected by exogenous application with FeNPs (T1). Untreated plant (**A**–CT) and plant treated with 6 nm FeNPs (**B**–T1) (×100). Uep = Upper epidermis; Pa = palisade tissue; Sp = Spongy tissue; Lep = lower epidermis; Xt = xylem tissue; Pht = phloem tissue; Mr = midrib region.

**Figure 5 plants-11-02599-f005:**
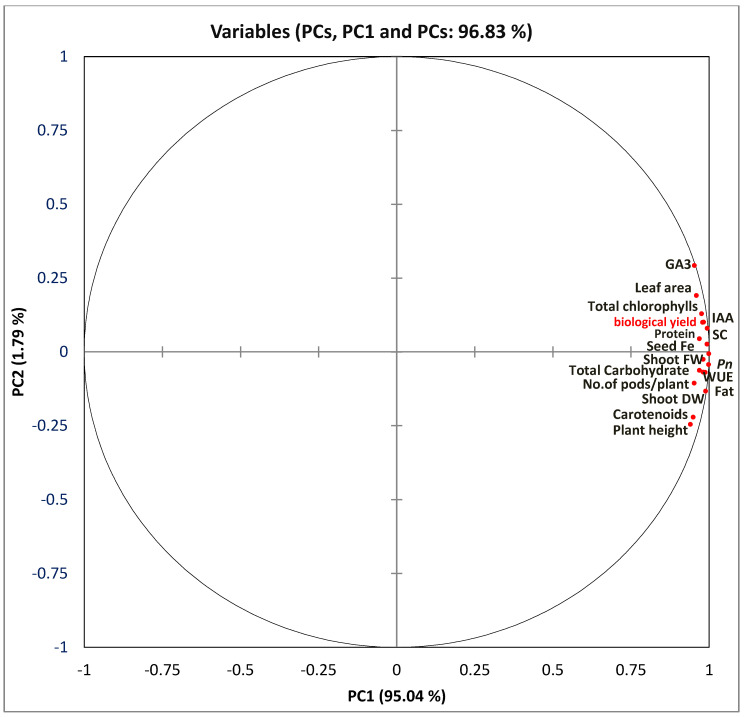
Principal component analysis of agro-physiological and biochemical properties of broad bean plants sprayed with the diverse form of iron fertilizers (nano, chelate, and sulfate).

**Figure 6 plants-11-02599-f006:**
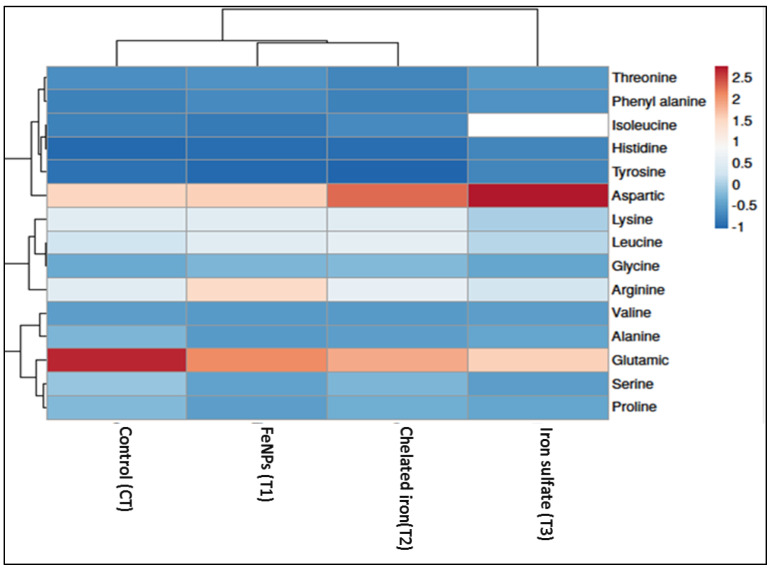
Correlation heat map between different amino acids in seeds of *Vicia faba* and different iron sources. Positive relationships are presented in red color and negative relationships in blue color. T1 = Synthesized nano-iron, T2 = Chelated iron, T3 = Iron sulfate, and CT = Control.

**Figure 7 plants-11-02599-f007:**
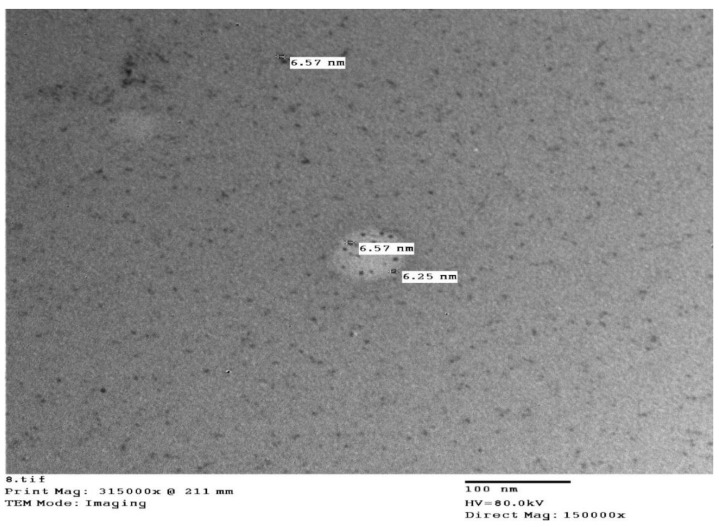
Synthesized magnetite iron nanoparticles under transmission electron microscopy (TEM).

**Table 1 plants-11-02599-t001:** Impacts of diverse iron sources (nano, chelate, and sulfate), as foliar spray on some growth attributes of *Vicia faba* plants.

Treatments	Plant Height (cm)	Leaf Area (cm^2^)	No. of Branches /Plant	Shoot FW (g/plant)	Shoot DW (g/plant)
**CT**	78.4 ± 3.22 ^d^	12.52 ± 1.25 ^c^	3.2 ± 0.21 ^d^	150.62 ± 5.23 ^d^	37.11 ± 1.23 ^cd^
**T1**	120.6 ± 6.55 ^a^	20.46 ± 3.01 ^a^	6.8 ± 0.51 ^a^	191.40 ± 4.1 ^a^	45.3 ± 3.2 ^a^
**T2**	106.3 ± 4.21 ^b^	15.33 ± 1.45 ^b^	5.1 ± 0.42 ^b^	175.33 ± 3.1 ^b^	41.8 ± 0.98 ^b^
**T3**	98.5 ± 2.55 ^c^	13.62 ± 2.34 ^c^	4.3 ± 0.15 ^c^	156.61 ± 3.7 ^c^	39.13 ± 0.25 ^c^

Values followed by the same letter (s) within each column did not significantly differ according to the Duncan multiple comparison test at the 5% level. Each value is the average of 6 replicates over two seasons. ±Values indicated to standard error (±SE). T1 = Synthesized nano-iron, T2 = Chelated iron, T3 = Iron sulfate, and CT = Control. FW= fresh weight and DW= Dry weight.

**Table 2 plants-11-02599-t002:** Impacts of diverse iron sources (nano, chelate, and sulfate), as foliar spray on photosynthetic pigments and photosynthesis apparatus of *Vicia faba* plants.

Treatments	Total Chlorophyll (mg.100 g^−1^ F.W)	Carotenoids (mg.100 g^−1^ F.W)	*Pn* (µmol m^–2^s^−1^)	SC (mmol m^–2^s^−1^)	WUE (μmol. mmol^−1^)
**CT**	86.49 ± 4.66 ^c^	22.19 ± 1.22 ^b^	8.41 ± 0.7 ^c^	0.21 ± 0.005 ^c^	9.11 ± 0.98 ^c^
**T1**	101.12 ± 5.01 ^a^	27.11 ± 2.02 ^a^	20.01 ± 0.8 ^a^	0.35 ± 0.007 ^a^	18.06 ± 1.08 ^a^
**T2**	90.45 ± 4.23 ^b^	24.79 ± 1.87 ^b^	14.15 ± 0.4 ^b^	0.29 ± 0.003 ^b^	12.09 ± 2.01 ^b^
**T3**	89.71 ± 3.99 ^c^	23.61 ± 3.11 ^b^	10.16 ± 0.2 ^c^	0.25 ± 0.009 ^b^	10.67 ± 0.790 ^c^

Values followed by the same letter (s) within each column did not significantly differ according to the Duncan multiple comparison test at the 5% level. Each value is the average of 6 replicates over two seasons. ±Values indicated to standard error (±SE). T1 = Synthesized nano-iron, T2 = Chelated iron, T3 = Iron sulfate, and Control = CT. FW = fresh weight, *Pn* = photosynthesis rate, SC = Stomatal Conductance, and WUE = Water Use Efficiency.

**Table 3 plants-11-02599-t003:** Impacts of different iron sources (nano, chelated, and sulfate), as foliar spray on seed’s nutrients content of *Vicia faba* plants.

Treatments	N (%)	P (%)	K (%)	Ca (%)	Fe (ppm)	Zn (ppm)	Mn (ppm)
**CT**	1.91 ± 0.11 ^c^	0.09 ± 0.03 ^d^	1.30 ± 0.33 ^d^	0.11 ± 0.001 ^b^	51.8 ± 5.11 ^d^	32.6 ± 2.32 ^d^	24.6 ± 1.54 ^b^
**T1**	3.85 ± 0.89 ^a^	0.39 ± 0.009 ^a^	1.76 ± 0.15 ^a^	0.45 ± 0.007 ^a^	128.4 ± 6.23 ^a^	67.5 ± 3.54 ^a^	32.1 ± 2.56 ^a^
**T2**	2.51 ± 0.21 ^b^	0.22 ± 0.002 ^b^	1.54 ± 0.32 ^b^	0.17 ± 0.001 ^b^	88.3 ± 4.11 ^b^	50.2 ± 4.21 ^b^	25.3 ± 2.89 ^b^
**T3**	2.04 ± 0.23 ^b^	0.13 ± 0.008 ^c^	1.42 ± 0.23 ^c^	0.12 ± 0.002 ^b^	68.7 ± 3.99 ^c^	41.8 ± 2.77 ^c^	25.8 ± 3.98 ^b^

Values followed by the same letter (s) within each column didn’t significantly differ according to the Duncan multiple comparison test at the 5% level. Each value is the average of 6 replicates over two seasons. ±Values indicated to standard error (±SE). T1 = Synthesized nano-iron, T2 = Chelated iron, T3 = Iron sulfate, and CT = Control.

**Table 4 plants-11-02599-t004:** Impacts of diverse iron sources (nano, chelated, and sulfate), as foliar spray on seed moisture and some metabolites contents.

Treatments	Total Carbohydrate (mg.g^−1^ DW)	Crude Protein (%)	Fat Content (%)	Moisture Content (%)
**CT**	493.29 ± 23.18 ^c^	26.30 ± 2.21 ^b^	1.55 ± 0.08 ^b^	7.30 ± 0.34 ^b^
**T1**	619.48 ± 34.12 ^a^	35.14 ± 3.11 ^a^	2.03 ± 0.12 ^a^	8.49 ± 0.51 ^a^
**T2**	546.73 ± 32.78 ^b^	30.25 ± 2.99 ^a^	1.74 ± 0.22 ^b^	7.65 ± 0.32 ^b^
**T3**	521.18 ± 44.02 ^b^	28.41 ± 3.01 ^b^	1.63 ± 0.43 ^b^	7.45 ± 0.22 ^b^

Values followed by the same letter (s) within each column didn’t significantly differ according to the Duncan multiple comparison test at the 5% level. Each value is the average of 6 replicates over two seasons. ±Values indicated to standard error (±SE). T1 = Synthesized nano-iron, T2 = Chelated iron, T3 = Iron sulfate, and CT = control. DW = Dry weight.

**Table 5 plants-11-02599-t005:** Impacts of diverse iron sources (nano, chelate, and sulfate), as foliar spray, on *Vicia faba* yield and its components.

Treatments	No. of Pods/Plant	No. of Seed/Pod	Harvest Index	Biological Yield (ton/ Fad)	Weight of 100 Seed (g)
**Control**	12.4 ± 1.23 ^c^	3.5 ± 0.21 ^b^	40.25 ± 2.34 ^c^	3.46 ± 0.23 ^a^	71.43 ± 4.33 ^c^
**T1**	23.8 ± 2.27 ^a^	5.21 ± 0.29 ^a^	84.07 ± 4.11 ^a^	5.5 ± 0.27 ^b^	90.21 ± 5.12 ^a^
**T2**	16.5 ± 2.12 ^b^	3.8 ± 0.15 ^b^	68.25 ± 6.24 ^b^	4.2 ± 7.01 ^b^	78.62 ± 3.11 ^b^
**T3**	13.2 ± 2.13 ^c^	3.6 ± 0.18 ^b^	45.33 ± 5.11 ^c^	3.41 ± 6.22 ^a^	75.48 ± 3.98 ^c^

Values followed by the same letter (s) within each column did not significantly differ according to the Duncan multiple comparison test at the 5% level. Each value is the average of 6 replicates. ±Values indicated to standard error (±SE). T1 = Synthesized nano-iron, T2 = Chelated iron, T3 = Iron sulfate, and CT = Control.

**Table 6 plants-11-02599-t006:** Counts and measurements in microns (µ) of certain histological characters in transverse sections through the first leaflet blade of the compound leaf developed on the median portion of the main stem of faba bean plant as affected by exogenous application with FeNPs (T1).

Histological Characters	Treatments		
	Control	(Fe-NPs) (6 nm in Size) (T1)	± % to Control (CT)
Midvein thickness	670	1070	+59.7
Lamina thickness	265	384.9	+45.2
Upper epidermis	22.5	26.6	+18.2
Lower epidermis	17.5	18.3	+4.5
Palisade tissue thickness	118.3	185	+56.3
Spongy tissue thickness	107.3	155	+44.4
Dimension of the midvein bundle:			
Length	380	555	+46.0
Width	640	970	+51.5
Phloem tissue thickness	170	260	+52.9
Xylem tissue thickness	210	295	+40.4
No. of xylem rows/midvein bundle	9	12	+33.3
No. of vessels/midvein bundle	83	100	+20.4
Vessel diameter	25	32.5	+30

**Table 7 plants-11-02599-t007:** The chemical and microbial properties of the applied compost.

Total Macro-Elements (%)				Total Micro-Elements (ppm)				O.M (%)	Organic-C (%)	E.C (dSm^−1^)	pH (1:5)	CEC cmol.kg^−1^
N	P	K	C/N	Fe	Mn	Cu	Zn					
1.82	1.29	1.25	14:1	1019	111	180	280	70	33.11	3.1	7.5	165
Total content of Bacteria				Phosphate dissolving Bacteria						Humidity		
2.5 × 10^7^				2.5 × 10^6^						25		

Each value in this table is the mean of 3 replicates. CEC = Cation Exchange Capacity, O.M = Organic Matter.

**Table 8 plants-11-02599-t008:** Physical and chemical properties of the experimental soil.

Particle Size Distribution				Texture Grade	E.C (dSm^−1^)	pH
Coarse Sand %	Fine Sand %	Silt %	Clay %			
63.16	30.28	4.9	1.67	Sand	0.86	7.8
Ava-Fe (ppm)		Ava-Zn (ppm)		Ava-N (ppm)	Ava-P (ppm)	Ava-K (ppm)
3.78		2.34		21.53	8.25	205.56
Cations in soil paste extract (me/l)				Anions in soil paste extract (me/l)		
Ca^++^	Mg^++^	Na^+^	K^+^	HCO_3_^−^	Cl^−^	SO_4_^−^
4.10	1.40	2.82	0.25	1.55	6.09	6.09

Each value in this table is the mean of 3 replicates.

## Data Availability

The data for this study are included in the main document.

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
