# Peer review of "Foliar Application of Different Iron Sources Improves Morpho-Physiological Traits and Nutritional Quality of Broad Bean Grown in Sandy Soil"

_plants, 2022, doi:10.3390/plants11192599_

Round 1

Reviewer 1 Report

My comments are attached as word document. 

Author Response

The authors would like to express their deep thanks to the reviewer for his/ her valuable comments that increase the quality of the current manuscript. Please open the attached files where was respond to comments and inquiries. 

Reviewer 2 Report

The paper describes the effect of different iron forms on the broad bean describing many of the morpho-physiological traits. The work can be considered for publication after addressing the following comments:

In the methodology for the synthesis of Fe3O4 magnetic nanoparticles (L519) authors refer to the previously published work with “slight modifications”. However, in the cited reference, only one source of iron ions is used (FeCl3*6H2O) while in the current work authors mention two. This might cause significant differences between the material synthesized in the cited work and in the current study. A lack of proper description of the process might also cause reproducibility problems.

In the methodology for the gas exchange parameter, the authors describe the usage of “five different leaves per treatment” (L558). This information should be more precise, were the leaves from the same plant? What was their weight?

Methodology for seed nutrient content (L565) lacks information about the number of seeds used, their weight, and the number of plants from which they were collected.

Introduction requires a description of the currently used forms of nano-fertilizers, please refer to https://doi.org/10.1007/978-981-32-9374-8_8

More characterization techniques are required to describe synthesized nanoparticles such as XRD (to confirm present phases) and zeta potential (to measure stability in water, since the material was used in the water suspension).

The comparison between previously studied iron nanoparticles and their effect would be more clear in the form of the table (with plant species, concentration, nanoparticle morphology, and observed effect).

A common phenomenon observed with nanoparticle application is an increase in ROS formation, which could be a good addition to the current work.

Author Response

The authors would like to express their deep thanks to the reviewer for his/ her valuable comments that increase the quality of the current manuscript. please open the attached files where was respond to comments and inquiries. 

Reviewer 3 Report

The current study entitled “Foliar Application of Different Iron Forms Improves Morpho-physiological Traits, and Nutritional Quality of Broad Bean Grown in Sandy Soil” is good. For a better understanding in-depth, it is need for time to work on this topic. Furthermore, achieving potential benefits by using current technology depends on extensive research work for more exploration. Although the experiment is well organized, I suggest a major revision due to the following deficiencies.

Major Concerns

Title

  • It is fine. However, I suggest changing the word iron form. As you can see, plants can uptake only one form of ion. You can write iron sources of application rather than forms.

Abstract

  • The systematic abstract is missing. Introduce the need for study in 1-2 lines.
  • Please give a clear-cut point problem source as a problem statement that is tackled in the current study.
  • Give a logical reason for selecting the current strategy, i.e., Foliar Application of Different Iron Forms.
  • Quantitative data is also essential to support your conclusion. I request that the authors carefully check and rewrite the results part of the abstract. Please provide a percentage increase or decrease in the result part.
  • Please provide a definitive conclusion withdrawn through research in a single line.
  • Please conclude with a statement that shows a knowledge gap covered, potential beneficiaries, and specific recommendations.
  • Give future perspective in a single line. At least declare one best result.
  • As per standard suggestions, please avoid using title words as keywords.

Introduction

  • Please follow the title in the introduction section, i.e., Foliar application of fertilizer, Different Iron Forms, Morpho-physiological Traits of Broad Bean, Nutritional Quality of Broad Bean, fertilizer management Sandy Soil, knowledge gap, hypothesis, and aims.
  • Also, provide a novelty statement at the end. What new things have authors done or correlated in this research compared to old ones?
  • Would you please give a single line about the knowledge gap your research has covered along with the SMART (specific, measurable, achievable, realistic, and time-specific) hypothesis statement?

Material and methods

  • Give the GPS location of the experimental site.
  • The most important thing in compost is cation exchange capacity. Please provide that.
  • How seeds were washed and cleaned. Either sterilization or seed priming done before sowing or not?
  • Give the reference for this “sulfuric acid and perchloric acid (3:1)”. Most di-acid mixture includes HNO3.

Results

  • I request that the authors they can provide Pearson correlation and parallel plots for a better understanding of the data.
  • Chord diagrams can also be made to clear the percentage contribution of each studied attribute.
  • Overall results are ok.

Discussion

  • Please provide a definite mechanism associated with the results. The discussion part is fragile.
  • Please incorporate at least 3-4 paragraphs showing the principal mechanism for which authors got such results.

Conclusion

  • The conclusion is so much descriptive. Please provide a conclusive conclusion.
  • Add the targeted beneficiary audience who will get benefit from this research.
  • Also, give clear-cut recommendations
  • Give future prospective regarding this research.

Author Response

(The authors gave the same response as above.)

Round 2

Reviewer 1 Report

1)    If protein content is measured, then why amino acid content was measured separately?

Discuss this point in discussion

2)    What is the relation of iron treatment and zinc content?

Where is it discussed in discussion?

3)    “The concentration of phytohormones (indole-3-Acetic Acid, gibberellic acid, and Ab- 151 scisic acid) in the leaf broad bean significantly affected the foliar application of Fe fertilizers 152 (Nano, chelate, and sulfate), as shown in Figure 1.” Which one is dependent and which one is independent factor?

Sentence is still wrong in manuscript

Author Response

Please see the attachment where the reply to the comments.

Reviewer 3 Report

Dear Authors

I am satisfied with the changes made in the manuscript.

Regards

Author Response

The authors would like to thank the Reviewer 3 for positive respond